# Lipid-Based Nanocarriers for Delivery of Neuroprotective Kynurenic Acid: Preparation, Characterization, and BBB Transport

**DOI:** 10.3390/ijms241814251

**Published:** 2023-09-18

**Authors:** Ádám Juhász, Ditta Ungor, Norbert Varga, Gábor Katona, György T. Balogh, Edit Csapó

**Affiliations:** 1Interdisciplinary Excellence Center, Department of Physical Chemistry and Materials Science, University of Szeged, Rerrich B. Sqr. 1, H-6720 Szeged, Hungary; vargano@chem.u-szeged.hu; 2MTA-SZTE Lendület “Momentum” Noble Metal Nanostructures Research Group, University of Szeged, Rerrich B. Sqr. 1, H-6720 Szeged, Hungary; ungord@chem.u-szeged.hu; 3Institute of Pharmaceutical Technology and Regulatory Affairs, Faculty of Pharmacy, University of Szeged, Eötvös Str. 6, H-6720 Szeged, Hungary; katona.gabor@szte.hu; 4Department of Pharmaceutical Chemistry, Semmelweis University, Hőgyes Endre út 9, H-1092 Budapest, Hungary; balogh.gyorgy@vbk.bme.hu; 5Department of Chemical and Environmental Process Engineering, Budapest University of Technology and Economics, Műegyetem Rakpart 3, H-1111 Budapest, Hungary

**Keywords:** liposome, kynurenic acid, blood–brain barrier, drug release

## Abstract

Encapsulation possibilities of an extensively investigated neuroprotective drug (kynurenic acid, KYNA) are studied via lipid-based nanocarriers to increase the blood–brain barrier (BBB) specific permeability. The outcomes of various preparation conditions such as stirring and sonication time, concentration of the lipid carriers and the drug, and the drug-to-lipid ratio are examined. Considering the experimentally determined encapsulation efficiency, hydrodynamic diameter, and ζ-potential values, the initial lipid and drug concentration as well as the stirring and sonication time of the preparation were optimized. The average hydrodynamic diameter of the prepared asolectin-(LIP) and water-soluble lipopolymer (WSLP)-based liposomes was found to be ca. 25 and 60 nm under physiological conditions. The physicochemical characterization of the colloidal carriers proves that the preparation of the drug-loaded liposomes was a successful process, and secondary interactions were indicated between the drug molecule and the polymer residues around the WSLP membrane. Dissolution profiles of the active molecule under physiological conditions were registered, and the release of the unformulated and encapsulated drug is very similar. In addition to this outcome, the in vitro polar brain lipid extract (porcine)-based permeability test proved the achievement of two- or fourfold higher BBB specific penetration and lipid membrane retention for KYNA in the liposomal carriers relative to the unformatted drug.

## 1. Introduction

Currently, more than 55 million people live with dementia worldwide, and Alzheimer’s disease represents nearly 70 percent of these cases. In this context, this neurocognitive disorder is the most common form of dementia. This syndrome places increasing social and economic burdens on the elderly, their relatives, nurses, and society. The fundamental trigger of major neurocognitive disorders remains unknown, but it is well known that the equilibrium between neurotoxic and neuroprotective agents profoundly impacts the function and survival of neurons [1,2,3]. Thus, the so-called kynurenine pathway (KP) has gained growing interest as its relationship to neurological conditions comes to be more and more apparent. The KP is the primary route for oxidative degradation of tryptophan in the liver in mammals [4]. Synthesis of the metabolites of this pathway is strongly controlled and may considerably vary under physiological and pathological circumstances. The kynurenine system includes several neuroactive compounds, which may be partly neurotoxic and partly neuroprotective. Experimental data consistently suggest that the shift of KP toward the neurotoxic branch or the relative or absolute absence of the neuroprotective kynurenic acid (KYNA) is an important factor in neurodegeneration [4,5]. For these reasons, it seems vital to target specific brain regions to maintain acceptable KYNA levels and this requires the development of novel and precise pharmacological tools that avoid possible adverse effects. Specific effects of KYNA are difficult to understand since its penetration over the blood–brain barrier (BBB) involves serious difficulty [6].

Besides the above-mentioned neurocognitive disorders, the managing of neuropathic pain remains a major challenge both for health providers and patients. Fortunately, KP also appears to be a potential moderator in the pathology of neuropathic pain [7,8]. Moreover, it was previously confirmed that KYNA can reduce the expression of a neuropeptide with migraine-inducing properties [9]. It should be noted that without a suitable carrier, KYNA cannot penetrate the BBB by interaction with plasma proteins alone [6]. However, there have been attempts to create nanosized carriers that allow therapeutic amounts of KYNA to cross the BBB. The results of these trials show that an encapsulated form of KYNA given to animals penetrated the BBB and caused electrophysiological changes in the central nervous system [10,11].

Considering all these findings together, the concept appears quite consistent that without a carefully designed drug nano-sized carrier, only a small percentage of KYNA reaches the target, and the excess drug may be responsible for certain side effects. It is well known the nanomaterials-based drug delivery tools offer numerous advantages such as reducing the amount of drug, delivering the drug in a targeted way, protecting the drug against chemical and enzymatic degradation, controlling drug release through physicochemical characteristics of the drugs and the carriers, and so on [12,13]. Many nanoscale or nanostructured materials have been tested as potential carriers of KYNA, but only a few articles have been reported in vitro [14,15], and there are even fewer reports about the results of in vivo experiments [16], although, our research group has proposed various colloidal reservoirs such as non-ionic surfactant-based micelles, protein-based core–shell nanoparticles [16], and layered double hydroxide nanosheets [15] as carriers for “small molecular” drugs such as KYNA [16] and ibuprofen [17]. For each listed colloidal carrier, there is some characteristic limitation which is specific to the given nano-object, so it is reasonable and useful to develop a newer and possibly more effective carrier system [18,19,20].

In view of all this, our research team recommend the use of liposome reservoirs consisting of a mixture of phospholipids and polyelectrolytes as potential KYNA carriers. These self-assembled (phospho)lipid-based drug carriers have been considered promising and versatile drug vesicles [21]. Liposomes are composed of a phospholipid layer (monolayer) and/or a concentric series of several bilayers (multilamellar), which enclose a central hydrophilic space with an aqueous inner phase [22]. The size of these colloid species ranges from 30 nm to the micrometer scale [23], with the lipid bilayer being typically 4–5 nm thick [24]. Compared with conventional drug delivery tools, liposomes show better properties, as well as site-targeting, prolonged release, defense of drugs from degradation, and lower toxic side effects [25].

The application of KYNA-based therapy could be a promising solution for neurocognitive disorders as well as migraine treatment [26]. However, due to the structure of this molecule, it can hardly cross the BBB, so it cannot be used directly for therapeutic objectives. A protein-based nanoparticle discussed in our early published manuscript [16] resulted in a US patent (US10857236B2) encapsulating KYNA in a core–shell type carrier and resulting in two-fold KYNA permeability through the BBB. Therefore, the aim of our study is to investigate the alternative possibility of increasing the therapeutic efficiency of this drug through liposomal carriers.

## 2. Results and Discussion

### 2.1. Optimization of the Preparation of the Liposomal Particles

To obtain samples with appropriate size (monodisperse, d < 100 nm) and maximum drug content, the commonly used thin-film hydration protocol was used, which was optimized, including the choice of the stirring/ultrasonication times as well as the applied lipid carrier and the initial drug concentrations and their optimal ratios.

During the first phase of experiments, we focused on optimizing the applied concentration of the lipid carrier components and the initial KYNA concentration (as it is represented in Appendix A). In the way, firstly the effect of drug concentration in aqueous solutions was studied according to the change of encapsulation efficiency (EE%). Then, the influence of different lipid concentrations was investigated for the same reason. The lipid concentrations given in the mg/mL dimension refer to the asolectin or lipopolymer content of the liposomal dispersion. The medium of this dispersion is the aqueous solution of the drug at the amount given in the mM dimension. As Figure 1A shows, the encapsulation efficiency (EE%) of the lipid-based carriers does not depend significantly on the starting drug concentration. In contrast, as can be seen in Figure 1B, the altering of the concentration of applied asolectin lipid linearly increased the amount of the encapsulated drug from ~70% to ~85%.

As the next step, as can be seen in Table 1, fixed stirring and six different sonication times were used to optimize the most appropriate procedure via monitoring the change of the size distribution, ζ-potential, and EE% values (visualized in Appendix A). Based on the measured data summarized in Table 1, 10 min stirring time and 30 min ultrasonication time (highlighted in italics and bold) were chosen for further studies.

Considering Figure 1 and Table 1, sample 4 showed the smallest average diameter (d_DLS_ = 23 ± 10 nm) with good kinetic stability (ζ-potential is −58 ± 3 mV) and high EE% (69.4%), so this preparation protocol was chosen for other characterization and penetration investigations. To address any significant differences in dependent variables, a one-way analysis of variance (ANOVA) followed by a post-hoc test was performed (d_DLS_ vs. ζ-potential ****, *p* < 0.0001; d_DLS_ vs. EE% ns, *p* > 0.05; ζ-potential vs. EE% ****, *p* < 0.0001). In addition to the preparation of asolectin-based carriers (LIP) with negative surface charge, a synthetically produced water-soluble lipopolymer was also prepared (WSLP) and used to formulate KYNA. This lipopolymer has positive surface charge because of the presence of a polyethyleneimine (PEI) part. For preparation of WSLP-based liposomes, the same preparation protocol was used as for LIP. Since we found an analogous linear relationship (as shown in the bar graph in Appendix A) between EE% and lipopolymer concentration in the case of WSLP, the amount of encapsulated drug was around 70% in the case of the chosen (0.5 mg cm^−3^) lipid concentration.

### 2.2. Physicochemical and Structural Characterization of the Colloidal Carriers

To obtain deeper information about the structure of the carriers and the structural properties of the formulations as well as to investigate the interaction between the carrier and the active ingredient, several studies were performed.

Average hydrodynamic diameters of the drug-loaded LIP- and WSLP-type carriers were determined by DLS technique. Based on the size distribution curves (Figure 2A), the drug-loaded LIP-type carrier has a smaller (d_DLS_ = 23 ± 10 nm) diameter, while the polymeric liposomal (WSLP) formulation is nearly twice as large (d_DLS_ = 59 ± 12 nm). This observation is in good agreement with an earlier published value [27]. When compared with the size of previously published cationic liposomes, it can be concluded that it is much more favorable, since their size in all cases exceeded 100 nm [28].

Thermal analysis (DSC) was carried out to confirm the successful encapsulation of the drug. The DSC results of liposomes suggest enhanced entrapment efficiency of KYNA in the lipid bilayer. The DSC results of the liposomal samples in Figure 2B indicate the presence of a smaller entrapped amount in the polymer/lipid bilayer in addition to the main encapsulated amount of the drug in the case of the KYNA/WSLP carrier (more detailed form is presented in Appendix A).

With the aim of examining the possible interaction between the carriers and the drug from a thermodynamic point of view, calorimetric (ITC) measurements were performed. The evolution of the calorimetric signal (dQ/dt) and molar enthalpy change (ΔH) as a function of the drug/carrier molar ratio are presented in Figure 3A,B. The outcomes of the calorimetric measurements suggest the existence of a weak interaction between the drug and the polymer-modified liposomal carrier.

To confirm this assumption, one and/or two-site binding models were used to model the experimental data and to describe the interaction between the donor (liposomes in the measuring cell) and the ligand (KYNA dosed from the syringe). As can be seen in Figure 3A, the model function presented by continuous green line could not finely describe the entalpogram of the LIP-type carrier, while a good fit can be performed in Figure 3B which shows the entalpogram and two-site binding model curve of the KYNA/WSPL system. The negative Gibbs free energy of KYNA binding to the polymer residues of the membrane structure of WSLP indicating the spontaneity of the process was observed at 25 °C. This value of −14.24 kJ mol^−1^ for the first and −3.43 kJ mol^−1^ for the second site of the two-site binding model indicates the secondary interactions between the drug molecule and the polymer residues around the surface of the liposome membrane.

The change of the streaming potential values for WSLP dispersions were registered as a function of concentration of lipid titrated constant amount of the negatively charged sodium dodecyl sulphate (SDS) and KYNA within the WSLP. The results are presented in Figure 4A,B. On one hand, the change of the streaming potential values from a negative to positive value strongly confirms the positively charged interface of the WSLP. The registered streaming potential curves at first glance show that the KYNA can neutralize more surface charges than the SDS.

The abscissa of the graphs in Figure 4 shows the mass of WSLP in the liposomal dispersion added to the surfactant (and drug) solution during the measurement. The measured streaming potential of the colloid system (U_str_/mV) is plotted versus the independent values (m/mg). In this way, knowing the parameters of the fitted Boltzmann equation (continuous green lines), it becomes possible to calculate the x coordinate of the model function at Ustr = 0 mV, which corresponds to the charge neutralization point. As can be seen in Figure 4, the regular sigmoidal profile of the charge titration curves is rather deformed; therefore, a modified version of the Boltzmann equation was used to evaluate the experimental data. The presence of attractive interactions between opposite charges is clear in the case of anionic surfactants, but a charge compensation arrangement is also possible, when positively charged amine groups of WSLP interact with the negatively charged carboxylate group of KYNA [29].

Comparing the results of the measurements, it can be concluded that the charge titration with KYNA (Q^σ^_obs_ = 13.3 ± 1.1 mmol_c_ mg^−1^) was able to detect the presence of more positive charges than the equivalently charged anionic surfactant (Q^σ^_obs_ = 8.7 ± 1.2 mmol_c_ mg^−1^). This result suggests that KYNA can penetrate the mixed phospholipid-polyelectrolyte membrane of WSLP, so that the full amount of drug is not used to compensate for surface charges. In other words, the discrepancy can be explained by the increasingly well-founded assumption that the interaction between lipids and drugs [30,31] or surfactants [32,33] is not only electrostatic, even in the case of well-defined charge conditions, but is also the result of dispersion and hydrophobic interactions. In this way, we must consider that in addition to the drug enclosed in the inner phase of the liposomes, the specific adsorption layer and the lipid membrane trapping of the drug determine the release profile and penetration of the active ingredient.

### 2.3. In Vitro Drug Release Studies of the Drug from Liposomes

Drug release kinetics are a crucial element of formulation design, as they are the main controlling aspect of the drug delivery of the carrier in vivo and the subsequent release of the free drug. Based on the data of the dissolution tests (Figure 5) for the liposomal drug carriers, nearly 86–88% of the encapsulated drug was released at the end of examined period (t = 210 min), while for the free drug, 92% of the initial quantity passed through the membrane. The data points of Figure 5 are fitted by well-known drug release kinetic models. Analytical solutions of first- and second-order chemical kinetic equations, empirical formulas including Weibull [34] and Korsmeyer–Peppas [35], and the pseudo-steady-state Higuchi equation [36] were tried to describe the experimental drug release profiles [37].

Among the experimental formulas, the Weibull equation was able to describe the experimental results most accurately, while among the differential equations the second-order rate equation was the most precise. To measure the similarity the rate of dissolution of the drug, the corresponding half-time (t_1/2_) data were compared. In the case of the non-formulated drug, the value t_1/2_ = 0.50 ± 0.03 h was found, while this value was t_1/2_ = 0.64 ± 0.06 h in the case of LIP- and 0.47 ± 0.03 h in the case of the WSLP-type carrier.

### 2.4. Blood-Brain Barrier Permeability Study of the Encapsulated Drug

The results of the BBB-PAMPA permeability test of the liposomal KYNA formulations are shown in Figure 6. Both liposomal formulations showed elevated, BBB-specific permeability (Pe), membrane retention (MR), and flux referring to pure KYNA (summarized in Appendix A). Based on the determined Pe/MR data (Figure 6A,B), it can be assumed that the multifold growth in permeability of KYNA in the case of liposomal formulations can be attributed to increased membrane retention. Regarding the two colloidal carriers, the reduced Pe and MR of WSLP compared with LIP is in large measure the effect of the two-fold larger hydrodynamic diameter resulting from the presence of PEI. The Pe values of the drug (Figure 6A) showed significantly higher values for the two liposomes compared with the unformulated KYNA (KYNA vs. LIP/KYNA ****, *p* < 0.0001; KYNA vs. WSLP/KYNA ****, *p* < 0.0001; LIP/KYNA vs. WSLP/KYNA ****, *p* < 0.0001). The results of a neurochemistry study demonstrate that under normal conditions, KYNA crosses the blood–brain barrier poorly [6]. The permeability of a previously published cationic liposomes was significantly higher (16.2 ± 2.2 × 10^−6^ cm∙s^−1^) compared with that of neutral liposomes (1.6 ± 0.4 × 10^−6^ cm∙s^−1^), or unformulated drug (1.3 ± 0.4 × 10^−6^ cm∙s^−1^) [28]. In our case, the LIP formulation showed the higher Pe (21.3 ± 0.8 × 10^−6^ cm∙s^−1^) while the “cationic” WSLP gave a smaller (11.9 ± 0.5 × 10^−6^ cm∙s^−1^) value.

That is, the nanosized liposomes greatly facilitated the penetration across the BBB, which is a mechanistically critical step for drug permeability. In accordance with this finding, the BBB-specific flux values of the drug (Figure 6C) also showed significantly higher values for the two liposomes compared with the unformulated KYNA. The increased flux of liposome-enclosed and -entrapped drug can be explained by of two effects. Namely, in addition to the permeability, the equilibrium solubility of the drug via the formulations also significantly increased compared with the pure KYNA, due to encapsulation. Comparing the flux of the liposomal carriers, a significant difference was observed. This difference may indicate the existence of an excess drug amount entrapped through an electrostatic attraction between the oppositely charged drug molecules and polymer residues, in the case of the WSLP formulation.

## 3. Materials and Methods

### 3.1. Materials

For preparation and characterization of the lipid-based carriers, the following materials were used: asolectin (25% phosphatidylcholine, Sigma, Budapest, Hungary), sodium dodecyl sulfate (SDS) (≥99.0%, Sigma), poly(ethylenimine) (PEI, 30% wt % aqueous solution, Tokyo Chemicals) (the mean molecular weight of PEI was determined in a previous work using dynamic light scattering (DLS) measurements as Mw ≈ 139 ± 1 kDa [24]), trimethylamine ((C_2_H_5_)_3_N, 99%, Sigma), cholesteryl chloroformate (C_28_H_45_ClO_2_, 95%, Sigma), chloroform (CHCl_3_, 99.9%, Molar, Halásztelek, Hungary), methanol (CH_3_OH, 99.9%, Molar), poly(allylamine hydrochloride) (PAH, Mw = 17,500 g/mol, Sigma), methylene chloride (CH_2_Cl_2_, anhydrous, >99.8%, Sigma), diethyl ether ((CH_3_CH_2_)_2_O, anhydrous, >99.7%, Sigma), acetone (CH_3_COCH_3_, >99%, Molar), and kynurenic acid (KYNA, C_10_H_7_NO_3_, >98%, Sigma). For adjustment of the pH, sodium phosphate monobasic monohydrate (NaH_2_PO_4_ × H_2_O; 99%; Sigma), sodium phosphate dibasic dodecahydrate (Na_2_HPO_4_ × 12 H_2_O; 98.5%; Sigma), sodium hydroxide (NaOH, 99.8%, Molar), hydrochloric acid (HCl, 37%, Molar), and sodium chloride (NaCl, 99.9%, Molar) were applied. All the chemicals were analytical grade and were used without further purification. In all cases, the stock solutions were freshly prepared using Milli-Q (Millipore, Milli-Q Integral3, Merck, Budapest, Hungary) ultrapure water (18.2 MΩ·cm at 25 °C).

### 3.2. Preparation of Asolectin-Based Nanocarriers Containing KYNA

The asolectin-based carriers (LIP) were prepared using our previously published preparation protocol [25,26], which was slightly modified for successful encapsulation of the present neuroactive drug molecule. Namely, 100 mg of asolectin was dissolved in 10 mL CHCl_3_:CH_3_OH/9:1 mixture (c_lipid_ = 10 mg cm^−3^). The organic solvent was removed by vacuum evaporation for 8–10 min at 50 °C to form a lipid film on the flask wall. The solid lipid film was hydrated in 20 mL of aqueous solution containing KYNA (cKYNA = 0.01 M) by continuous magnetic stirring at 800 rpm for 20 min. To achieve optimal size and maximum encapsulation efficiency, the sample was further treated by a half-hour sonication (37 kHz) and continuous stirring at 800 rpm for 20 min. Finally, non-encapsulated KYNA excess was removed by gel chromatography. The scheme of the preparation is presented in Appendix A. The water-bloated hydrophilic matrix structured dextran gel (Sephadex^®^ G50 fine, Sigma) as the stationary phase was added into an Eppendorf-filter (0.5 mL capacity) and centrifuged for 5 min at 13,000 rpm for compression. The separation of the excess KYNA was performed using ultracentrifugation at 7000 rpm for 5 min.

### 3.3. Preparation of Lipopolymer-Based Nanocarriers Containing KYNA

The water-soluble lipopolymer-based carriers (WSLP) were fabricated based on our previously published preparation protocol [24], which was slightly modified for successful encapsulation of the present neuroactive molecule. Namely, 10 mL PEI solution (30% *m*/*v*) was mixed with 10 mL ice-cold CH_2_Cl_2_ and 100 µL trimethylamine. For the next step, 1.0 g of cholesteryl chloroformate was added to 5 mL of chilled CH_2_Cl_2_ and this solution was slowly mixed with the PEI solution. The mixture was stirred at 0 °C at 800 rpm for 12 h, resulting in a pale-yellow solid product. The scheme of the reaction is presented in Appendix A. The synthesized WSLP was purified by the following steps. First, the product was completely dried by rotary evaporation (IKA, RV 3 evaporator), then, the powder was dissolved into 50 mL of 0.1 m HCl, and extraction was repeated three times using 50 mL CH_2_Cl_2_. For the removal of the larger aggregates, vacuum filtration was applied using standard P3 glass and paper filters with 16–40 and 3–5 µm pore size, respectively. The filtered liquid was concentrated by rotary evaporation (50 °C, 2 h), then the WSLP was precipitated by acetone and the filtration was repeated as described above. Finally, the solid WSLP was obtained by freeze drying (Christ Alpha, 1 2 LD dryer) for 1 day. Due to the surface-active property of the lipid, for encapsulation the appropriate amount of WSLP and KYNA was simply mixed in Milli-Q water and incubated for 20 min at room temperature. WSLP in an aqueous environment behaves like a surfactant and when its concentration reaches a characteristic value (critical micelle concentration, cmc) it forms micelles in aqueous solutions thanks to its amphiphilic features. In an earlier published study, cmc of 0.35 ± 0.03 mg mL^−1^ was obtained by the fluorescent dye method [27,38,39].

### 3.4. Methods for Characterization

#### 3.4.1. Structural Characterization of the KYNA-Containing Carriers

Size distribution curves, average hydrodynamic diameter (d_DLS_), polydispersity index (PDI), and Zeta-potential (ζ-potential) values were determined via dynamic light scattering (DLS) method, using a HORIBA SZ-100 NanoParticle Analyzer (Retsch Technology GmbH, Haan, Germany). Carrying out the measurements, the applied methods and conditions were the same as in the case of previously presented liposome-based carrier systems [40]. For the measure of the enclosed amount of KYNA in the liposome and the released extent of the drug from the liposomal carriers, the absorbance spectra of the aqueous KYNA solutions were recorded on a Shimadzu UV-1800 UV spectrophotometer using a 1 cm quartz cuvette in the range of 200–800 nm. Differential scanning calorimetry (DSC) studies were carried out using a Mettler-Toledo 822e calorimeter. The materials were heated from 25 to 500 °C at a 5 °C/min heating rate.

#### 3.4.2. Isothermal Titration Calorimetry (ITC) Based Investigation

Calorimetric measurements were performed with a VP-ITC microcalorimeter from MicroCal Inc. (Northampton, MA, USA). The sample cell (1.416 mL in volume) was typically filled with the liposome suspension and the reference cell with the corresponding liposome-free buffer solution. The first aliquot of 10 μL followed by 26 aliquots of 10 μL of drug solution (pH 4.1 ± 0.1) were injected stepwise with 400 s interval into the working cell filled with the liposome suspension. The corresponding reference blank experiments were also performed, namely titration of drug-free buffer in the liposome suspension and titration of drug solution in liposome-free buffer. To avoid the presence of bubbles, all samples were degassed for 10 min shortly before starting the measurements. The sample cell was constantly stirred at a rate of 240 rpm, and the measurements were performed at 25 °C.

#### 3.4.3. Streaming Potential Measurements by Particle Charge Detector (PCD) Device

The surface charge of polymer-modified lipids can be crucial for the encapsulation of the active ingredient, so we investigated this characteristic in this type of drug carriers. A PCD-04 particle charge detector (Mütek Analytic GmbH, Herrsching, Germany) was applied to determine quantitatively the different charge neutralization points of the liposome-based colloids [41,42,43]. The specific surface charge of the WSLP was determined using an anionic surfactant (SDS). During this process, 10 mL of SDS solution (0.1 mM) was placed in the PCD, and 10 mL of aqueous liposomal dispersion (c = 0.025 mg/mL) was added in 0.5 mL portions. The streaming potential values were recorded 20 s after every addition of WSLP. The acquired result was examined and fitted taking into regard the fact that the standard sigmoidal character of charge titration curves is strongly altered in these cases. For this purpose, an altered version of the sigmoidal Boltzman equation [44] was used to describe the experimental data and to determine the charge neutralization point and thus the observed specific surface charge of the WSLP per unit mass (Q^σ^_obs_/mmol mg^−1^). After that, we repeated the PCD tests using the KYNA (c = 0.1 mM) instead of the anionic surfactant.

#### 3.4.4. In Vitro Dissolution and Release Studies

For the determine of the dissolved amount of KYNA (as a function of time) from LIP- and WSLP-type carriers, a Hanson vertical diffusion cell (VDC) was used. The vertical assembly of the diffusion cell provides a cylinder-shaped sample holder with 61 mm × 9 mm diameters and 4 mL sample volume. The KYNA-filled LIP and WSLP samples in the internal tank of VDC were separated from the outer PBS buffer (pH = 7.4, c_NaCl_ = 0.15 M) by a semipermeable cellulose membrane (cut-off: 12–14 kDa). The buffer media dispersed, and drug-loaded liposomes were continuously stirred at 800 rpm while the membrane-separated PBS was flooded through a sample loop with a peristaltic pump. The silicon tubes constructed a sample loop, connecting the VDC, the peristaltic pump, and a flow-through cuvette of the spectrophotometer. Thereby, the absorbance change of the dissolved KYNA was recorded almost in real time, and by thermostatic control of the water bath circulation, the measurements were carried out at 37 °C. Absorbance of drug molecules at 332 nm is exchangeable to concentration dimension via an adequate calibration process (the error of the calculated concentrations was less than ±2%); in this way the concentration of the free drug at an optional time could be calculated.

#### 3.4.5. Permeability Studies

In an earlier published study [45], a self-developed parallel artificial membrane permeability assay (PAMPA) system was applied to determine effective BBB-specific permeability of KYNA from reference solution and formulas in a comparison study. The filter donor plate (Multiscreen™-IP, MAIPN4510, pore size 0.45 µm; Millipore, Merck Ltd., Budapest, Hungary) was coated with 5 µL of lipid solution (16 mg brain polar lipid extract + 8 mg cholesterol dissolved in 600 µL dodecane). Then, the donor plate was fitted into the acceptor plate (Multiscreen Acceptor Plate, MSSACCEPTOR; Millipore, Merck Ltd., Budapest, Hungary) containing 300 μL of PBS solution (pH = 7.4), and 150 μL of the donor solution was put on the membrane of the donor plate. The donor plate was covered with a sheet of wet tissue paper and a plate lid to avoid evaporation of the solvent. The sandwich system was incubated at 37 °C for 4 h (Heidolph Titramax 1000, Heidolph Instruments, Schwabach, Germany), followed by separation of the PAMPA sandwich plates and the determination of concentrations of KYNA in the acceptor solutions by HPLC method in six parallel measurements.

The determination of KYNA concentration was performed with an Agilent 1260 HPLC (Agilent Technologies, Santa Clara, CA, USA). A Kinetex^®^ EVO C18 column (5 µm, 150 mm × 4.6 mm (Phenomenex, Torrance, CA, USA)) was used as stationary phase. The mobile phases consisted of 0.02 M KH_2_PO_4_ phosphate buffer adjusted to pH = 8.0 with 0.1 M sodium hydroxide (A), and methanol (B). An isocratic elution was performed for 5 min with 85–15% A-B eluent composition. Separation was performed at 30 °C with 1 mL min^−1^ flow rate, and 10 µL of the samples was injected to determine the KYNA concentration at 323 nm using a UV–VIS diode array detector. Data were evaluated using ChemStation B.04.03 software (Agilent Technologies, Santa Clara, CA, USA). The regression coefficient (R^2^) of the calibration curve was 0.999 in the concentration range 5–500 μg mL^−1^. The determined limits of detection (LOD) and quantification (LOQ) of KYNA were 302 ppm and 917 ppm, respectively.

The effective permeability (*P_e_*), membrane retention (MR), and flux (Flux) of the samples were calculated using Equations (1)–(3):(1)$$P_{e}=-\frac{2.303\cdot V_{A}}{A\left(t-\tau_{SS}\right)}\cdot \log\left[1-\frac{c_{A}\left(t\right)}{S}\right]$$
(2)$$MR=1-\frac{c_{D}\left(t\right)}{c_{D}\left(0\right)}-\frac{V_{A}c_{A}\left(t\right)}{V_{D}c_{D}\left(0\right)}$$
(3)$$Flux=P_{e}\cdot S$$
where *P_e_* is the effective permeability coefficient (cm/s), *A* is the filter area (0.24 cm^2^), *V_D_* and *V_A_* are the volumes in the donor (0.15 cm^3^) and acceptor phase (0.3 cm^3^), *t* is the incubation time (s), *τ_SS_* is the time to reach the steady state (s), *C_A_*(*t*) is the concentration of the compound in the acceptor phase at time point *t* (mol/cm^3^), *c_D_*(*t*) is the concentration of the compound in the donor phase at time point *t* (mol/cm^3^), *c_D_*(0) is the concentration of the compound in the donor phase at time point zero (mol/cm^3^), and *S* is the solubility of KYNA in the donor phase determined by previously presented method [41].

## 4. Conclusions

Preparation conditions and parameters affecting the potential medical application of neuroactive KYNA-containing liposomal carriers were studied in this research. The optimal initial concentration of the drug and the lipid was determined as c_KYNA_ = 1.0 mM and c_LIPID_ = 0.5 mg/mL. Under these conditions, the average hydrodynamic diameters of the liposomes can be found between 25 and 60 nm; that size can be taken up by the entrances of endothelial cells. With the applied concentration ratios and resulting diameters, the maximum accomplished EE% was around 70%. To compare the rate of dissolution of the drug, the corresponding half-time (t_1/2_) data were calculated. For KYNA, the value of t_1/2_ = 0.50 ± 0.03 h was found, while this value was t_1/2_ = 0.64 ± 0.06 h in the case of LIP- and 0.47 ± 0.03 h in the case of the WSLP-type carrier. Based on the permeability studies, it can be stated that the liposomal nanocarriers carriers did not show significant improvement in drug retention compared with the free drug, while BBB-specific permeability (Pe), membrane retention (MR), and flux of both the LIP- and WSLP-type carrier were found to results in multifold favorable values compared with the unformulated KYNA.

## Figures and Tables

**Figure 1 ijms-24-14251-f001:**
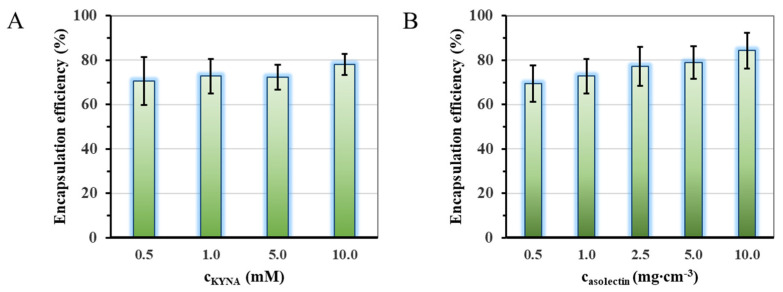
Change of the encapsulation efficiency of the lipid-based carriers as a function of the applied drug, *p* < 0.092 (**A**) and lipid, *p* < 0.028 (**B**) concentration. (A: c_KYNA_ = 1.0 mM; B: c_asolectin_ = 1 mg cm^−3^).

**Figure 2 ijms-24-14251-f002:**
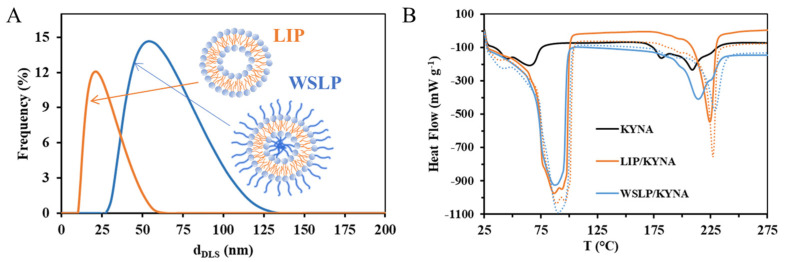
Average hydrodynamic diameter distribution curves of the KYNA-loaded liposomal carriers (**A**). Differential scanning calorimetry (DSC) curves of pure drug (KYNA as black line) and the liposomal (LIP/KYNA carrier as orange line and WSLP/KYNA carrier as blue line) formulations, where dotted lines represent thermograms of drug-free carries (**B**).

**Figure 3 ijms-24-14251-f003:**
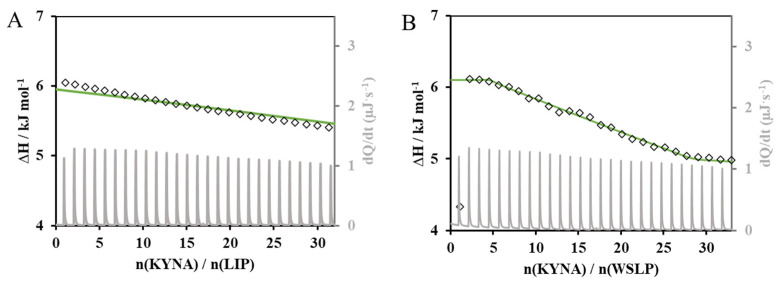
Calorimetric signal (dQ/dt as grey line), entalpogram (molar enthalpy change (ΔH) as a function of the drug/carrier molar ration; as empty diamonds) and two-site binding model curve (green line) for the KYNA/LIP (**A**) (c_LIP_ = 0.05 mg mL^−1^ and c_KYNA_ = 10.0 mM) and KYNA/WSPL (**B**) (c_WSLP_ = 0.5 mg mL^−1^ and c_KYNA_ = 4.0 mM) systems.

**Figure 4 ijms-24-14251-f004:**
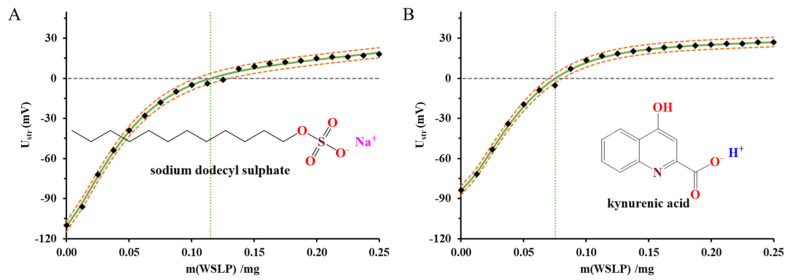
Changing of the streaming potential (mV) of WSLP dispersion as a function of concentration of dosed liposome (c_WSLP_ = 0.025 mg mL^−1^) at constant anionic surfactant concentration (c_SDS_ = 0.1 mM) (**A**) and at constant drug concentration (c_KYNA_ = 0.1 mM) (**B**); black squares symbolize the measured data, continuous green lines represent the calculated functions, and dashed orange lines indicate the width of the confidence interval.

**Figure 5 ijms-24-14251-f005:**
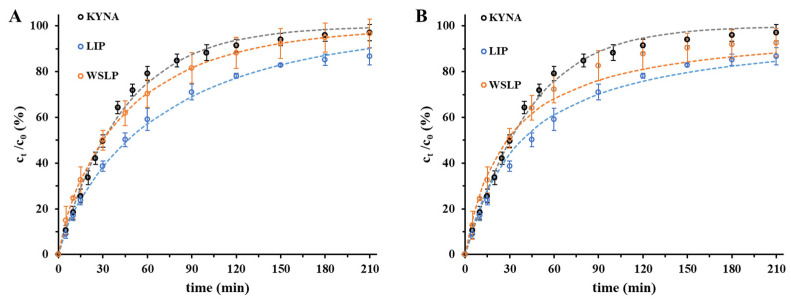
Dissolution data (circles) and Weibull equation evaluated release curves (dashed lines) of the KYNA and the drug-containing liposomal carriers (**A**). Dissolution data (circles) and first order rate equation evaluated release curves (dashed lines) of the KYNA and the drug-containing liposomal carriers (**B**) at pH = 7.4 and 37 °C.

**Figure 6 ijms-24-14251-f006:**
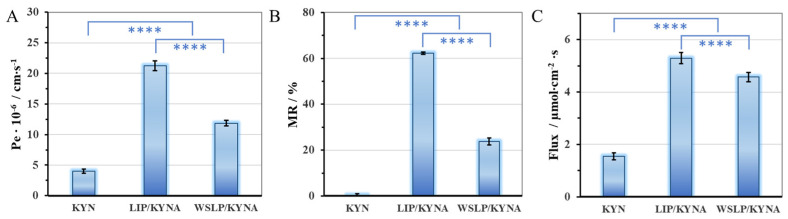
BBB-specific permeability (**A**), membrane retention (**B**), and flux (**C**) values of BBB-PAMPA permeability study of LIP- and WSLP-type carriers compared to initial KYNA, calculated by Equation (3). Statistical analysis was performed using one-way analysis of variance (ANOVA) followed by Bonferroni multiple comparison post-hoc test. Formulations were tested in triplicate, two independent experiments were conducted, and data are expressed as mean ± standard deviation (SD). The difference was considered statistically significant at *p* < 0.05; ****, *p* ≤ 0.0001.

**Table 1 ijms-24-14251-t001:** Average hydrodynamic diameter, ζ-potential, and encapsulation efficiency (%) values of the LIP/KYNA samples using different sonication times for treatment (c_LIP_ = 0.5 mg cm^−3^, c_KYNA_ = 1.0 mM).

Sample	Sonication Time (min)	d_DLS_ (nm)	ζ-Potential (mV)	EE%
1.	0	152 ± 34	−67 ± 4	58.0
2.	10	34 ± 18	−62 ± 5	55.4
3.	20	42 ± 21	−62 ± 3	70.0
** *4.* **	** *30* **	** *23 ± 10* **	** *−58 ± 3* **	** *69.4* **
5.	40	36 ± 10	−53 ± 4	48.8
6.	50	44 ± 13	−61 ± 4	54.0

## Data Availability

The datasets used and evaluated during the presented study are available from Ádám Juhász (juhaszad@chem.u-szeged.hu) upon reasonable request.

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
