# Peer review of "Lipid-Based Nanocarriers for Delivery of Neuroprotective Kynurenic Acid: Preparation, Characterization, and BBB Transport"

_ijms, 2023, doi:10.3390/ijms241814251_

Round 1

Reviewer 1 Report

Re: IJMS 2604494 “Lipid-based nanocarriers for delivery of neuroprotective kynurenic acid: preparation, characterization and BBB transport” by Juhasz et al, submitted to IJMS

This article describes two lipid-based nanocarriers for kynurenic acid (KYNA), an important neuroprotective agent. The first nanocarrier is a liposome (LIP) composed of azolectin: a commercially-available mixture of (principally) phosphatidylcholine, phosphatidylethanolamine, and phosphatidylinositol.  The second nanocarrier is a self-assembly of water soluble lipopolymers (WSLP) composed of a cholesteryl-polyethylenimine conjugate prepared in the authors’ laboratory.  The authors employ various methods to characterize certain aspects of the KYNA-nanocarrier composites and their ability to permeate a lipomembrane model barrier.

Overall, the background and rationale for this study is well presented, as are the various studies undertaken.  And the figures are well-done.  The English language usage, however, requires a thorough going-over by a proficient editor.  Now to more substantive concerns.

Fig 1 shows encapsulation efficiency (EE%) results for the azolectin liposomes.  No comparable studies of the EE% for the WSLP are presented.  Why not?

Table 1 is described (pg 3, line 107) as showing “six different stirring and sonication times” when in fact only a single stirring time was used, versus six different sonication times. Please correct the description.

Fig 2A compares size distributions for the LIP versus the WSLP nanocarriers.  The individual PEI polymers forming the polymeric surface coating of the WSLP are 135 kDa in mass, equating to approx. 3000 monomer units (assuming a repeat unit molar mass of 43 Da), i.e. quite lengthy.  Since smaller size equates to more ready permeation, the authors should comment on the utility of examining WSLP constructed with shorter PEI chains.

Fig 2B reports DSC results for KYNA-loaded nanocarriers of both types. The description states simply that the results “suggest enhanced entrapment efficiency of KYNA in the lipid bilayer” (pg 4 line 133) and (page 4 lines 134-136) “indicate the presence of a smaller entrapped amount in the polymer/lipid bilayer besides the main encapsulated amount of the drug in the case of KYNA/WSLP carrier.” I am confused.  Do the authors mean that KYNA is present in both the aqueous interior of the nanocarriers AND the lipidic bilayer?  And that there is some difference in the amount of KYNA in the lipidic bilayer of the azolectin versus WSLP?  Plus, how did they arrive at this conclusion based on the DSC endotherms displayed?  The large endothermic events around 100 C are clearly water-related, so I assume it is the endotherms in the region of 225 C that are in question.  Yet one is broad and the other narrow, so it is the area under the endotherm that matters.  Obviously, further explanation of how they reached their conclusions is necessary.

Fig 6 is described as a blood-brain barrier permeability study of the nanocarrier encapsulated drug.  The blood-brain barrier consists of the layer of epithelial cells lining the blood vessels supplying the brain. The permeability barrier employed by the authors is more akin to the old black lipid film technique which coated a pinhole with a mixture of phosphatidylcholine (or other lipid) with dodecane, forming a single bilayer thick barrier through which permeation was measured.  So this is not a blood brain barrier or even a very good model thereof.  That is not to say such studies are not useful.  But it is not a blood brain barrier and it is hyperbolic to describe it as such.

Ignoring these criticisms, would the authors not agree that the reduced permeability and membrane retention of WSLP versus the azolectin is in large measure the result of the greater WSLP size and the presence of the PEI coating which produces a steric barrier, rather than merely (page 7 lines 243-244) “electrostatic attraction between the oppositely charged drug molecules and polymer residues in the case of the WSLP formulation.”

Overall, I conclude that this article makes a significant contribution, but must undergo significant revision before publication.

Trivialities:

Azolectin or asolectin ?  Choose one. (page 3, line 102)

Pg 6 line 208, Fig 5 NOT 3

Fig 2 caption, calorimetry NOT colorimetry

Please have the manuscript edited by a proficent english speaker / writer.  Although mostly fine, there are multiple points of necessary improvement.

Reviewer 2 Report

In the manuscript titled “Lipid-based nanocarriers for delivery of neuroprotective kynurenic acid: preparation, characterization and BBB transport”,  the authors have employed surface-modified liposomes to encapsulate the neuroprotective agent kynurenic acid, aiming to enhance its permeability through the blood-brain barrier. To achieve this, a comprehensive research effort has been undertaken, though acceptance is contingent upon addressing the following significant revisions:

  1. The basis of the concentrations mentioned in the abstract, in relation to solute and solvent, should be clarified to ensure understanding.
  2. The optimization procedure, crucial to the synthesis of these nanocarriers, should be supplemented with detailed data in the form of supplemental information.
  3. Statistical significance in EE% concerning drug and lipid concentrations, if observed in Figure 1A and B, should be explicitly indicated in the revised figure.
  4. Exploring the potential for generating a predictive numerical model based on the data from Table 1 would be insightful.
  5. Adequate statistical analysis should be included to address any significant differences in dependent variables mentioned in Table 1.
  6. Consider presenting Table 1 as a 3-dimensional figure for improved visualization.
  7. Explanation for the observed trend in d(DLS) and EE% with respect to sonication time, including a comparison with relevant published studies, is needed.
  8. For comprehensive confirmation of successful drug encapsulation, thermograms of LIP and WSLP without the drug should be provided alongside the DSC thermogram in Figure 2B.
  9. Statistical analysis and identification of any significant differences in Figure 6A and B are recommended.
  10. The inclusion of a schematic reaction pathway to synthesize LIP and WSLP would facilitate understanding of the process.
  11. A significant omission in the manuscript is the absence of microscopic images of synthesized LIP and WSLP. SEM/TEM images would greatly enhance the morphological characterization.
  12. The discussion section should feature a paragraph that contextualizes this work within the realm of similar contemporary research and explains how it addresses existing gaps in knowledge.

The manuscript generally exhibits a good quality of English language; however, there are instances where certain sentences appear lengthy and could benefit from being rephrased in a more succinct manner. 

Round 2

Reviewer 1 Report

The authors have addressed my concerns, including any shortcomings of English language usage. I find the manuscript now acceptable for publication.

Reviewer 2 Report

The manuscript could be accepted in its current form. the authors have clearly addressed all the concerns.